# Cryo-electron tomography reveals structural insights into the membrane remodeling mode of dynamin-like EHD filaments

Arthur A. Melo[1,2] ✉, Thiemo Sprink[1,3], Jeffrey K. Noel[1], Elena Vázquez-Sarandeses[1,2], Chris van Hoorn ®[1], Saif Mohd[1,2], Justus Loerke ®[4], Christian M. T. Spahn[4] & Oliver Daumke ®[1,2] ✉

Eps15-homology domain containing proteins (EHDs) are eukaryotic, dynamin-related ATPases involved in cellular membrane trafficking. They oligomerize on membranes into filaments that induce membrane tubulation. While EHD crystal structures in open and closed conformations were previously reported, little structural information is available for the membrane-bound oligomeric form. Consequently, mechanistic insights into the membrane remodeling mechanism have remained sparse. Here, by using cryo-electron tomography and subtomogram averaging, we determined structures of nucleotide-bound EHD4 filaments on membrane tubes of various diameters at an average resolution of 7.6 Å. Assembly of EHD4 is mediated via interfaces in the G-domain and the helical domain. The oligomerized EHD4 structure resembles the closed conformation, where the tips of the helical domains protrude into the membrane. The variation in filament geometry and tube radius suggests a spontaneous filament curvature of approximately 1/70 nm$^{-1}$. Combining the available structural and functional data, we suggest a model for EHD-mediated membrane remodeling.

Eps15-homology domain-containing proteins (EHDs) comprise a conserved dynamin-related ATPase family in eukaryotes[1]. Mammals contain four closely related EHD homologues[2], while only a single member is present in *Drosophila* (termed PAST-1) and *C. elegans* (Rme-1). Rme-1 mediates the release of cargo receptors from the endocytic recycling compartment[3,4]. A similar function was demonstrated for mammalian EHD1 and EHD3[5,6], which also function in the formation of ciliary vesicles[7,8]. Furthermore, a conserved tethering complex, including EHD1 was shown to coordinate vesicle scission and fusion at sorting endosomes[9]. EHD2 assemble at the neck of caveolae in ring-like oligomers[10–15] that control cellular fatty acid uptake[12,16]. EHD4/Pincher mediates macropinocytosis required for retrograde endosomal Trk signaling[17,18]. EHD4 also recruits EHD1 to sorting endosomes via hetero-

dimerization[19]. Most recently, a role of an EHD4 complex in the trafficking of vascular endothelial cadherin (VE-cadherin) during angiogenesis was revealed[20].

EHDs harbor a dynamin-related GTPase (G-) domain that binds to adenine rather than guanine nucleotides[21,22]. EHDs tubulate negatively-charged liposomes in an ATP-dependent manner by the formation of ring-shaped and helical filaments on the remodeled membranes[22–24]. Similar to other dynamin superfamily members, oligomerization on membranes stimulates nucleotide hydrolysis[22,25]. In reconstitution experiments, ATP hydrolysis in EHD1 induces bulges in tubular membrane templates, leading to membrane scission[25].

The crystal structure of EHD2 in the presence of a non-hydrolyzable ATP analogue revealed a dynamin-related extended

[1]Max-Delbrück-Center for Molecular Medicine in the Helmholtz Association, Structural Biology, Robert-Rössle-Straße 10, Berlin, Germany. [2]Freie Universität Berlin, Institute of Chemistry and Biochemistry, Takustraße 6, Berlin, Germany. [3]Cryo-Electron Microscopy Core Facility, Charité - Universitätsmedizin Berlin at the MDC, Robert-Rössle-Straße 10, Berlin, Germany. [4]Institut für Medizinische Physik und Biophysik, Charité - Universitätsmedizin Berlin, Berlin, Germany. ✉e-mail: arthur.melo@ucsf.edu; oliver.daumke@mdc-berlin.de

G-domain that mediates stable dimerization via an EHD family-specific interface (interface-1) (Supplementary Fig. 1a, b)[22]. Residues at the N- and C-terminal ends of the G-domain form a composite helical domain. In the crystal structure of the EHD2 dimer, the two helical domains protrude in parallel away from the G-domains. This orientation was termed the 'closed' conformation of EHDs. The tip of the helical domain was shown to constitute the primary membrane-binding site[23].

The C-terminal EH domains interact with linear peptide sequences containing Asn-Pro-Phe (NPF) motifs[26] that are present in binding partners, such as MICAL-L1[27,28], Rabenosyn-5[9,29], EHBP1[30] and PACSIN1/2[13,15,31]. In the EHD2 dimeric structure, the EH domains bind back to a Gly-Pro-Phe (GPF) motif in the opposing monomer (Supplementary Fig. 1b). In this orientation, the C-terminal tails of the EH domains are positioned in the nucleotide-binding site and block the G-interface. The EH domains may thereby auto-inhibit EHD assembly[22].

The crystal structure of an N-terminal deletion variant of EHD4 in the presence of a nonhydrolysable ATP analogue revealed a 50° rotation of the helical domains compared to the EHD2 crystal structure (Supplementary Fig. 1b)[32]. In this conformation, the two helical domains in the dimer protrude away from each other. This arrangement was termed the 'open' conformation of EHDs. Spectroscopic experiments indicated that EHD2 binds in the open conformation to flat membrane bilayers[33]. Furthermore, the EH domains were displaced from the G-domain in the EHD4 structure, suggesting that the open conformation is not compatible with EH-domain mediated auto-inhibition[32].

An N-terminal sequence stretch folds back into a conserved hydrophobic pocket of the G-domain in the EHD2 structure[23] (Supplementary Fig. 1). In the presence of membranes, the N-terminal residues were shown to insert into the lipid bilayer[23]. In turn, a flexible loop at the periphery of the G-domain, the 'KPF loop', inserts into the hydrophobic pocket of the G-domain and serves as an oligomerization interface[32]. Deletion of N-terminal residues is therefore expected to stabilize the KPF loop in the G-domain pocket and promote oligomerization. Accordingly, enhanced membrane recruitment and oligomerization was observed for EHD2 and EHD4 variants lacking their N-terminal residues[23,32].

Based on the two available crystal structures, a nucleotide-driven activation model of EHDs has been proposed[32]. However, since membranes were lacking in any of the reported structures, the detailed conformation of EHDs on membranes and, consequently, their membrane remodeling and oligomerization mode have remained unknown.

In the present work, we reconstitute an N-terminally truncated EHD4 variant on tubular membrane templates and determine these structures by cryo-electron tomography (cryo-ET) and subtomogram averaging (STA). We thereby clarify the oligomerization mode, reveal how EHD4 interacts with membranes, demonstrate how the EHD4 oligomer adapts to various membrane curvatures and propose a model for EHD-mediated membrane remodeling.

## Results

### Cryo-ET and subtomogram averaging to elucidate the structure of membrane-bound EHD4

To understand the molecular mechanisms of EHDs assembly on membranes, EHD4-coated membrane tubes were reconstituted in vitro. To this end, we used a previously described N-terminal deletion construct of mouse EHD4 (EHD4$^{\Delta N}$, corresponding to amino acids 22–541) that forms a regular protein coat on membranes and displays enhanced membrane recruitment when overexpressed in eukaryotic cells compared to full-length EHD4[32]. In contrast to EHD4$^{\Delta N}$, we did not obtain soluble full-length EHD4 with the same expression protocol.

As reported earlier[32] and similar to EHD1[25] and EHD2[23], EHD4$^{\Delta N}$ remodeled liposomes in an ATP-dependent fashion (Supplementary Fig. 2a) and showed liposome-stimulated ATPase activity

(Supplementary Fig. 2b). Accordingly, our in vitro reconstitutions were done in the presence of the nonhydrolysable ATP analogue, adenylyl-imidodiphosphate (AMPPNP). Liposomes containing 50% Folch extract from bovine brain, 40% phosphatidylethanolamine and 10% cholesterol were chosen as template since they reproducibly yielded densely coated lipid tubes. These tubes were highly heterogeneous, with luminal diameters ranging between 30 to 100 nm (Fig. 1a) and a variety of different shapes (Supplementary Fig. 2a and Supplementary Movie 1). The protein coat had a thickness of ~12 nm (Fig. 1a) without and of ~16 nm including the lipid bilayer (Fig. 1b, c). In contrast to EHD4$^{\Delta N}$, EHD2 and an EHD2 variant lacking its N-terminus deformed liposomes into membrane tubes of much smaller diameters (Supplementary Fig. 3).

To determine the structure of the EHD4 coat, we used cryo-ET and reference-free subtomogram averaging (STA) due to the heterogeneity of tube diameters and helical families (Fig. 1b). For this, we collected 56 tilt-series using dose-symmetric tilt scheme, from −60° to 60° with 3° of increment. We divided the data into two half-datasets, which were processed independently (Supplementary Fig. 4a). The structures were compared by Fourier Shell Correlation (FSC) and averaged to generate a final structure from 23,813 subtomograms. The resulting cryo-electron microscopy (cryo-EM) density map had an average resolution of 7.6 Å and 8.5 Å, as determined by FSC and FSC-independent analyses, respectively (Supplementary Table 1, Supplementary Fig. 4). Consistently, α-helices of EHD4 could be discerned in the best-resolved parts of the map (Fig. 1c, Supplementary Fig. 5, Supplementary Movie 2). The overall architecture of membrane-bound EHD4 was highly similar to the closed EHD2 dimer (see below), which allowed us to confidently place and orient the G-, helical and EH domains in the density maps by fitting them as rigid bodies (Fig. 1d). This yielded a pseudo-atomic structure of the assembled EHD4 coat, which was further refined using a flexible fitting strategy (see Methods and Supplementary Movie 2).

### Structural determinants of the EHD4 filaments

EHD4 formed right-handed helical filaments wrapping around the lipid tubes (Fig. 1e). The EH domains are located furthest from the membrane, the G-domains at the center and the helical domains closest to the membrane (Figs. 1e, 2a). The unit particle used for subtomogram averaging contained three parallel filaments of oligomeric EHD4$^{\Delta N}$. The center filament (F$_i$) is composed of three EHD4 homodimers, and the adjacent filaments (F$_{i+1}$ and F$_{i-1}$) of three EHD4 monomers each (Figs. 1d, e and 2a). As in the previously reported crystal structures, dimerization of EHD4 is mediated by helix α6 in the G-domain. It forms a two-fold symmetric interface, to which we refer as interface-1 (Fig. 2b, Supplementary Movie 3). A point mutation of a conserved tryptophan in this interface (W238A) renders the protein insoluble (Fig. 3)[22].

Each EHD4 dimer interacts in the filament through two additional interfaces (Fig. 2b, Supplementary Movie 3). Interface-2 is formed between the helical domain of one protomer and the G-domain of the adjacent protomer along the filament. It involves the KPF loop in the G-domain and helices α8 and α12 of the adjacent helical domain. The density of the KPF loop at the periphery of the G-domain dimer is consistent with the open EHD4 crystal structure (Fig. 2b, Supplementary Fig. 5). Point mutations in this loop were previously shown to disrupt oligomerization (Fig. 3)[32].

Interface-3 is formed between G-domains of two adjacent dimers across the filament (Fig. 2b) and corresponds to the archetypal G-interface that is conserved in all members of the dynamin family. Dimerization via this interface induces nucleotide hydrolysis in dynamin-related proteins[34]. Highly conserved residues in the switch I, switch II, the EHD signature motif and the N-terminal part of α6 are involved in this contact close to the active site. Accordingly, point mutations in these elements in EHD2 were previously shown to abrogate stimulated ATP hydrolysis (Fig. 3)[22].

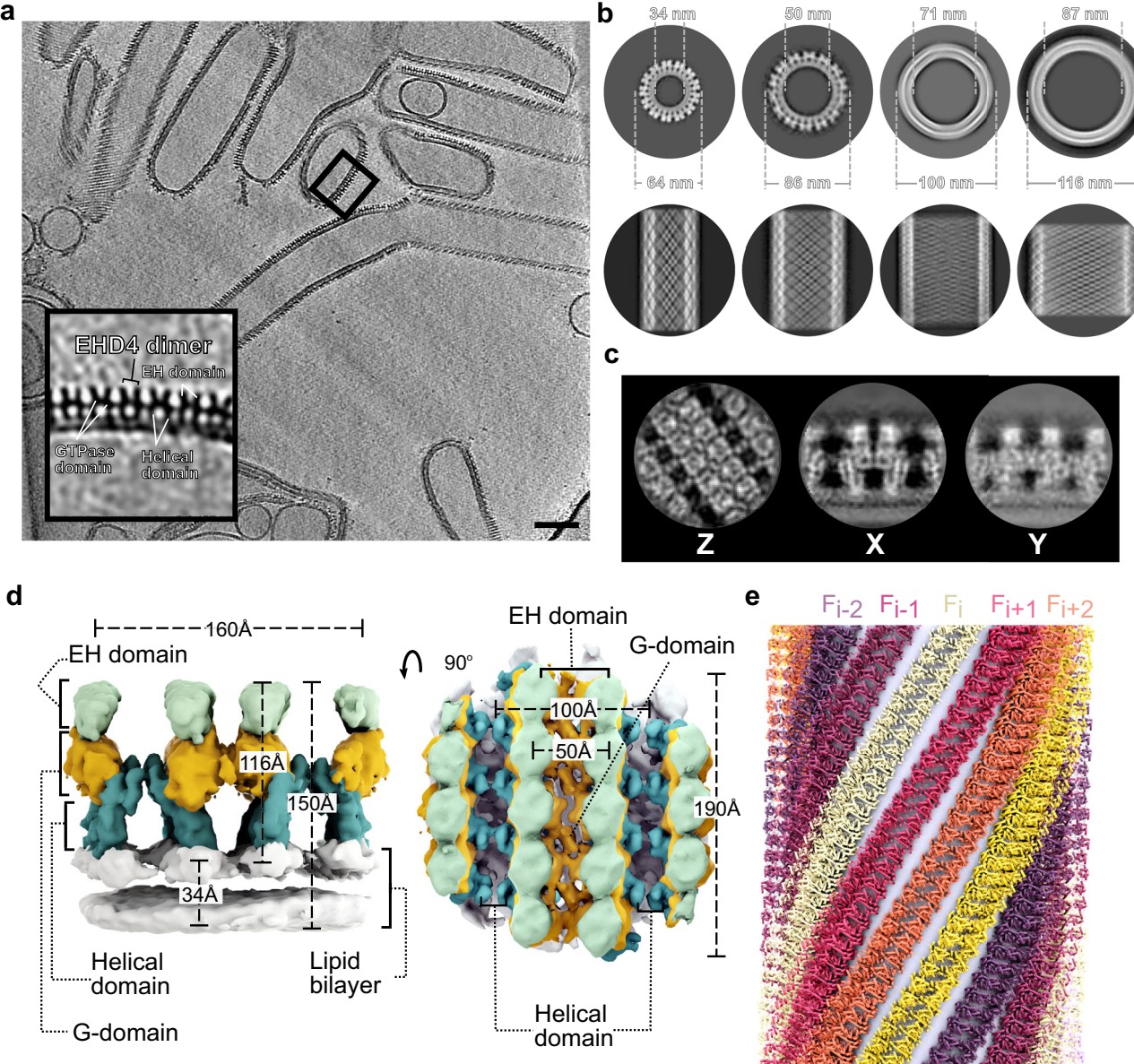

**Fig. 1 | Structure determination of membrane-bound EHD4. a** Tomogram reconstruction of EHD4-covered membrane tubes. The boxed area is magnified in the left bottom corner. Scale bar is 100 nm. **b** Projection of subtomogram averages of individual tubes with different diameters and helical families. Each tube was individually cropped and averaged using Dynamo. Top: Cross section of a tube. Tubes are ordered from the smallest to biggest diameters. Bottom: Z projection of EHD4 tubes that show different helical families according to different diameters. **c** Projections of the membrane-bound map using 23,813 particles. **d** Subtomogram average of the membrane-bound EHD4^ΔN complexed with AMPPNP at 7.6 Å resolution. Domain organization of EHD4 in the filaments in two orientations is shown. EH domains (light green), G-domains (orange), helical domains (teal) and the lipid bilayer (white) are colored individually. **e** Reconstructed EHD4 right-handed helical filaments wrapping around a membrane tube. Each filament is indicated in a different color.

In the EHD2 crystal structure, the EH domains bind back to the opposing G-domains[22]. Similar to this closed conformation, the EH domains were also located on top of the G-domain in our membrane-bound structure (Supplementary Fig. 6a, b). However, compared to the EHD2 structure, they were 15 Å shifted away to the periphery (Supplementary Fig. 6c) and stabilized by contacts with EH domains of adjacent dimers (Supplementary Fig. 6b, d). Accordingly, in this orientation, the C-terminal auto-inhibitory tail may not reach into the active site so that the G-interface can be formed.

### Membrane-binding mode of EHD4
We compared the EHD4 structure to the reported crystal structures of EHD2 and EHD4. Membrane-bound EHD4 adopted the closed conformation of the helical domain, akin to the reported EHD2

conformation (Fig. 4a). In this conformation, the long central helix α8 from each monomer protrudes towards the membrane. The root-mean-square deviation of Cα atoms between the closed crystal structure and the membrane-bound EHD4 conformation, excluding the EH domains, is only 2.4 Å for the monomer, attesting to the high similarity of the two structures. Thus, surprisingly, the closed, supposedly auto-inhibited EHD2 dimeric crystal structure represents a good model for the membrane-bound EHD4 state determined in this study.

The membrane bilayer was well-defined in our map and had a thickness of about 45 Å (Fig. 4b). The outer and inner layers were clearly separated. The strong density at the periphery of the bilayer likely corresponds to the phosphate headgroups, which have greater signal in comparison to the fatty acid tails, as seen already in other cryo-ET structures of membrane-bound protein structures[35,36]. Due to

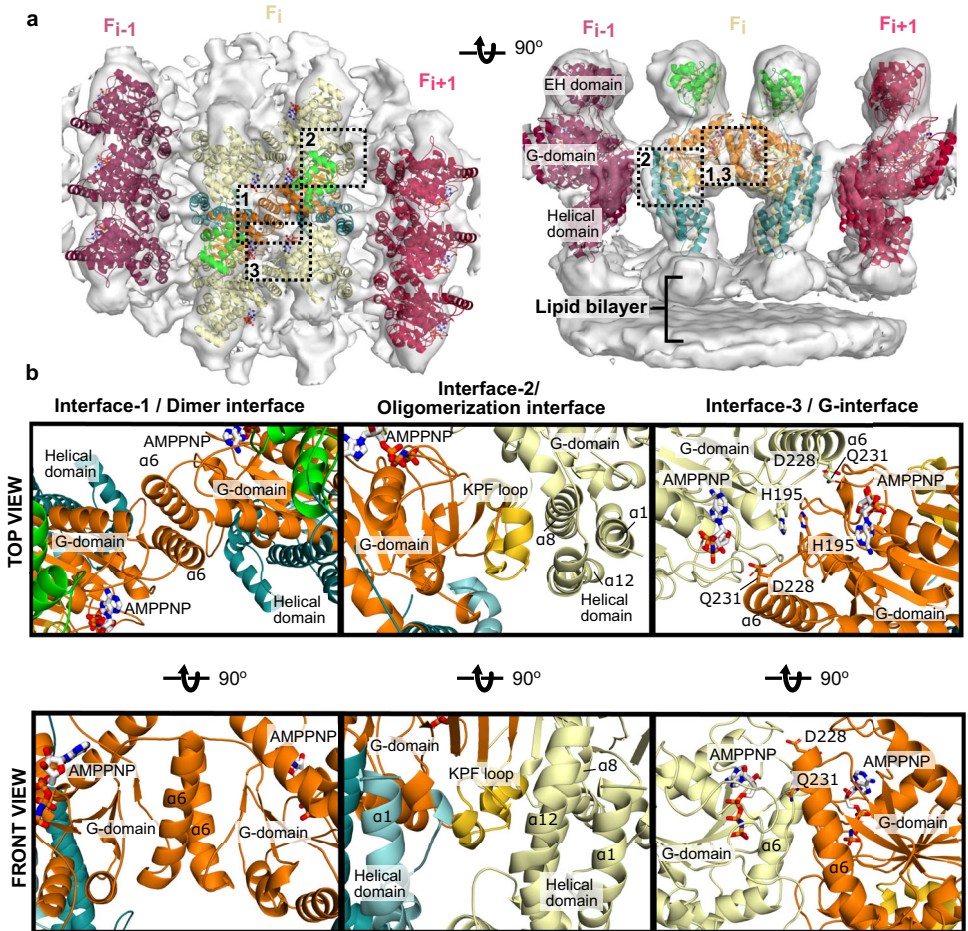

**Fig. 2 | Architecture of the EHD4 filament. a** Asymmetric unit of the cryo-ET reconstruction. In the central EHD4 dimer, the domains are colored according to the domain architecture. The cryo-ET density is indicated as grey surface. **b** The filament is formed by three contact sites. Interface-1 comprises a dimer interface in the G-domain, interface-2 is built by contacts between the G-domain and helical domain whereas interface-3 represents the G-interface and may involve the nucleotide.

the lack of bilayer structures and models, we refrained from adding a bilayer model to our map.

By electron paramagnetic spin resonance experiments and mutagenesis, EHD2 residues at the tip of the helical domain were shown to insert into the membrane (Fig. 3, Fig. 4c)[23]. These findings are consistent with the structure of membrane-bound EHD4. Thus, helices α8 and α9 in the helical domain engage with the membrane bilayer by inserting hydrophobic residues at the connecting loop into the membrane outer leaflet (Fig. 4b, c). We employed the helical domains from the EHD4 crystal structure to analyze the side chain positions in relation to the membrane outer layer. In the membrane-bound structure, the Cα atoms of residues K331 and R332 are 1.5-3 Å above the outer leaflet of the bilayer, whereas residues E328, K330 were 0.5–2.5 Å below (Fig. 4c). Residues N323, M324 and F325 at the α8-α9 connecting loop deeply insert into the membrane, with their Cα atom 4, 6, and 8 Å below the membrane density, respectively (Fig. 4c). Charged residues in helix α9, such as K327, K330 and E333 were not inserted into the membrane but were close enough for interaction with the polar lipid head groups (Fig. 4c). Residues in the membrane binding region are highly conserved amongst EHD paralogues and across different species (Fig. 4d). However, each EHD paralogue has a unique combination of membrane-interacting residues (Fig. 4c, d). Thus, EHD proteins bind to membranes through charged residues at helices α8 and α9, which likely confer lipid specificity, and conserved hydrophobic residues at the connecting loop[22,37]. Notably, EHD4 membrane interaction resulted in a local bending of the outer membrane monolayer (Fig. 4b, insets).

**EHD4 oligomers assemble on membranes of different curvature**

EHD4[ΔN]-coated tubes had a wide range of radii (Fig. 5a). Cryo-ET and STA allowed us to probe the architecture of the EHD4[ΔN] coat on individual tubes, and therefore, the geometry of individual filaments to be discerned. The EHD4[ΔN] coat was governed by a set of related helical families, which varied in pitch, rise, and subunits per turn (Figs. 1b and 5b). By tracing the position of the filaments with the refined orientations in the tomograms, we measured the helical angle of several filaments as a function of the underlying tube radius (Fig. 5a, b, Supplementary Fig. 7a, b).

The helical angle, i.e. the deviation of the helical filament from a simple ring around the tube, tended to increase as the tube radius decreased. This relationship is expected for an elastic filament with a spontaneous curvature less than the curvature of the wrapped tube. Similar behavior is seen for highly constricted dynamin-coated tubes[38,39]. A best fit suggests that the spontaneous curvature of the AMPPNP-bound-EHD4[ΔN] filament is 1/68 nm$^{-1}$ (Fig. 5a, bottom). Fitting to more complicated elastic models suggests that the twist stiffness of the filament is significantly weaker than the curvature stiffness and that the angle of each EHD4[ΔN] dimer relative to the tube axis plays little to no role in determining the helical angle (Supplementary Fig. 7c).

Since it requires energy to generate membrane curvature, the ability of EHD4[ΔN] to produce membrane tubes smaller than its spontaneous curvature suggest some additional factor(s) promoting curvature. An obvious candidate is the membrane interaction of EHD4[ΔN]. An alternative is the interaction of helices α1 and α12 from neighboring

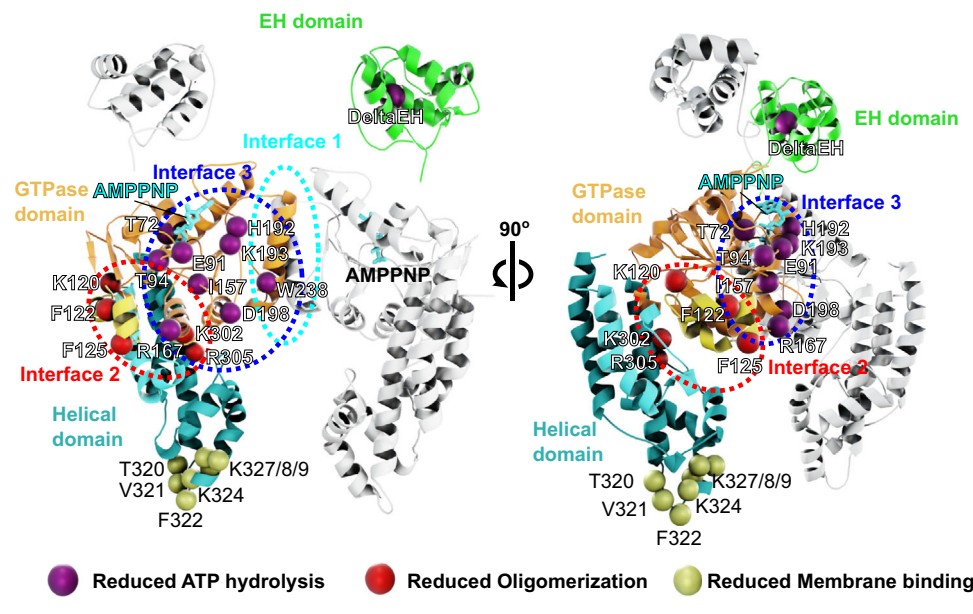

| Mutation | Protein mutated | Effect | Reference | EHD4 residue |
|---|---|---|---|---|
| **GTPase domain (interfaces 1 and 3)** | | | | |
| W238A | EHD2 | Results in insoluble protein | Daumke et al., 2007 | W241 |
| T72A | EHD2 | ATP-binding deficient | Daumke et al., 2007 | T75 |
| | | Cytosolic distribution *in vivo* Incapable of binding membranes and localizing to caveolae | Moren et al., 2012 | |
| T94A | EHD2 | Reduced liposome-stimulated ATPase activity | Daumke et al., 2007 | T97 |
| | | Altered caveolae structure | Moren et al., 2012 | |
| I157Q | EHD2 | Increased ATPase activity | Daumke et al., 2007 | I160 |
| H192D, K193D R167E, D198R | EHD2 | Liposome-stimulated ATPase activity deficient | Daumke et al., 2007 | H195, K196 R170, D201 |
| H192D, K193D | EHD2 | Distorted caveolae structure | Moren et al., 2012 | H195, D201 |
| **GTPase domain (KPF loop) / helical domain (interface 2)** | | | | |
| F122A/F128A | EHD2 | ATPase activity deficient and unable to localize to caveolae | Daumke et al., 2007 Moren et al., 2012 | F125/F131 |
| K120N | EHD2 | Caveolae mislocalization | Moren et al., 2012 | K123 |
| F125A | EHD4 | Membrane remodeling deficient and cytoplasmic distribution | Melo et al., 2017 | F122 (EHD2) |
| K302A/R305A | EHD4 | Decreased membrane oligomerization. Liposome-stimulated ATPase activity deficient. | Melo et al., 2017 | K299/R302 (EHD2) |
| **Helical domain** | | | | |
| F322A, K324D, K327D, K328D K329 | EHD2 | Cytoplasmic distribution *in vivo*. Reduced membrane binding *in vitro* | Daumke et al., 2007 | F325, K327 K330, K331 R332 |
| T320, V321, F322, K328, | EHD2 | Direct contacts to membrane | Shah et al., 2014 | S323, V324, F325, K331, |
| **EH domain** | | | | |
| Delta EH domain | EHD2 | Lack of ATPase stimulated activity and membrane remodeling | Moren et al., 2012 Daumke et al., 2007 | |
| F122A/Delta EH domain | EHD2 | Caveolae mislocalization | Moren et al., 2012 | |

**Fig. 3 | The membrane-bound EHD4 structure explains the effects of mutations.** Previously reported point mutants of EHD2 and EHD4 were plotted on the EHD4 structure. The effects of the mutations and the relevant literature is listed. All mutagenesis data are consistent with our oligomerized EHD4 structure.

EHD4 monomers (interface-4), which mediates the packing of neighboring filaments into a continuous coat (Fig. 5c). Interface-4 shows variation as the radius of the tube varies, and preferences toward a particular orientation would influence the underlying tube radius. While EHD4 forms ordered oligomers on membrane tubes with radii ranging from 15–70 nm (Fig. 5a), it can also bind to very highly curved tubes (e.g. radius <10 nm), but apparently cannot form ordered oligomers (Fig. 5d).

## Discussion
Recent advances in cryo-EM have facilitated the structural analysis of membrane-bound protein scaffolds. Helical reconstructions requiring highly homogeneous samples (reviewed in[40]) have allowed, amongst others, medium to high-resolution structural elucidation of the acetylcholine receptor[41], BAR domain proteins[42,43], endosomal sorting complexes required for transport (ESCRT)[36], light-dependent proto-chlorophyllide oxidoreductase (LPOR)[35] and dynamin[44] assembled on membrane tubes. Structures of highly heterogeneous membrane-bound protein coats cannot be determined by helical reconstructions. However, recent advances in image processing have facilitated the structure solution of such specimens by cryo-ET analysis combined with subtomogram averaging. Although the average resolution of such reconstruction is often lower compared to those derived from single-particle cryo-EM analyses (see Supplementary Table 2), pseudo-atomic models can be obtained by fitting available higher-resolution crystal structures into the density. Examples for cryo-ET structures include the membrane-bound COPI[45] and COPII coats[46], the N-BAR protein Bin1 bound to membranes[47] and the retromer coat[48,49]. EHD4 samples

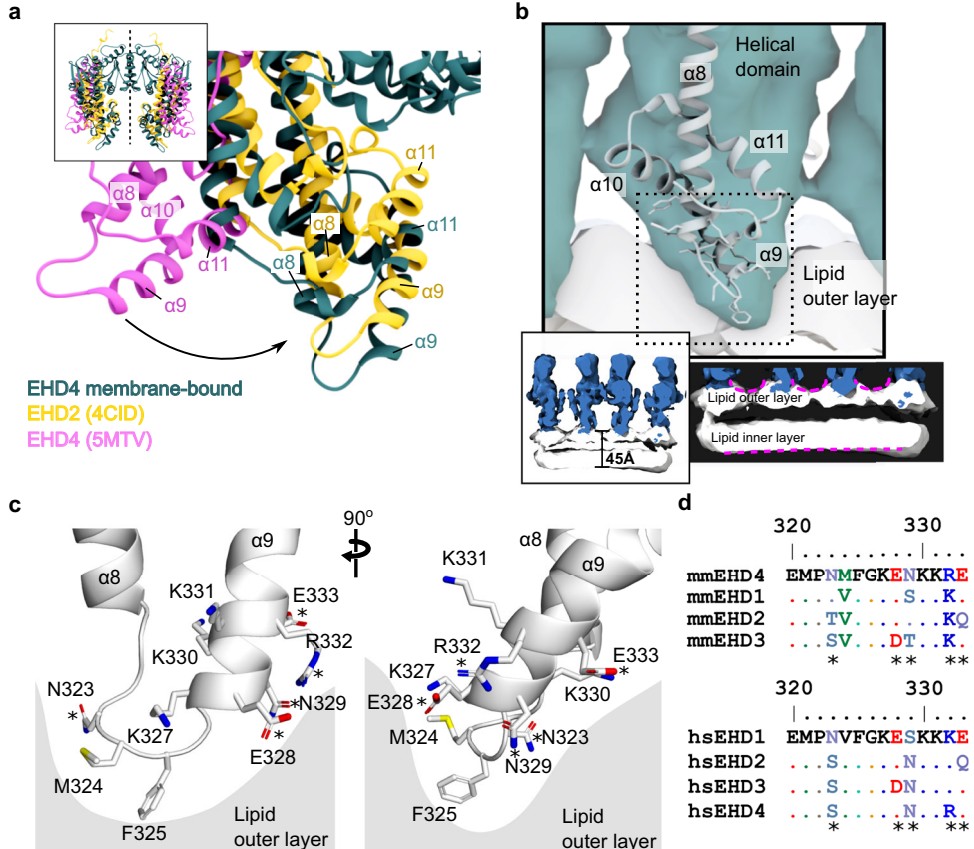

**Fig. 4 | The membrane-binding mode of EHD4. a** The cryo-ET model of EHD4 (teal) was superimposed with the G-domain on the crystal structures of EHD2 (yellow; pdb code 4CID) and EHD4 (pink; pdb code 5MTV). The comparison reveals that the membrane-bound structure adopts a closed conformation akin to EHD2. **b** Membrane-binding site of EHD4 inserts into the lipid outer layer. The two lower panels show how the membrane interaction of the EHD4 filaments induces buckling of the lipid outer layer, as indicated by the magenta dotted line. **c** Membrane-binding site of EHD4 based on the fittings shown in **a** and **b**. Side chains were modelled based on the EHD4 crystal structure. **d** Sequence alignments of EHD proteins of mouse (mm - *Mus musculus*) and human (hs – *Homo sapiens*) reveals a high conservation of the membrane binding site. Conserved residues in all 4 EHD proteins are shown in the sequence alignment as dots (.). Residues that differ in EHDs are highlighted (*).

bound to membrane tubes were highly heterogeneous[32], necessitating the use of cryo-ET for structural analysis. By tracking individual tubes from the refined subtomogram averages, this analysis allowed us not only to determine the structure of EHD4 within one filament but also to determine the EHD4 filament structures on various membrane curvatures. Our analysis has important implications for understanding how the ATPase cycle of EHDs is coupled to membrane recruitment, filament assembly and disassembly and how EHD4 generates membrane curvature. A resulting working model for the ATPase-dependent membrane cycle is outlined in the following.

Previous X-ray crystallographic analyses identified two conformations of EHDs. In the reported EHD2 crystal structure[22], the protein adopts a closed conformation, whereas the crystal structure of EHD4 features an open conformation[32] (Supplementary Fig. 1). Spectroscopic studies suggest that EHDs are recruited to flat bilayers in an open conformation[33]. Since the membrane interaction in the open conformation involves mostly polar interactions in the helical domain[32], EHDs may be in a rapid exchange with the cytosol. The G-domain is close towards the membrane in the open conformation so that the N-terminus can switch from its hydrophobic G-domain pocket into the membrane bilayer (Fig. 6a). The release of the N-terminus allows the KPF loop to enter the hydrophobic pocket to create oligomerization interface-2. Furthermore, our previous crystallographic study on EHD4 indicated that the G-interface (interface-3 in this manuscript) cannot be formed between EHD4 dimers in the open conformation due to steric constraints[32].

The transition of the open to the closed conformation, as observed in our study in the membrane-bound form, appears to be driven by the assembly of EHD oligomers on curved membranes (Fig. 6a, Supplementary Movie 4). By bilayer coupling[50,51], the insertion of the hydrophobic helical tip region into the membrane is expected to generate membrane buckling, in line with the undulating appearance of the outer membrane layer of the EHD4-coated tubes in our cryo-ET reconstructions (Fig. 4b) and previous molecular dynamics simulations[25]. Similar to FYVE and ENTH domains[52], the EHD membrane-binding site is composed of charged residues and a hydrophobic membrane-penetrating protrusion. In turn, curved membranes may facilitate the insertion of hydrophobic residues and therefore promote the transition from the open to the closed conformation in the EHD filament.

Upon initial curvature generation by this wedging mechanism, EHD filaments then assemble into ring-like or helical oligomers via interfaces-2 and -3 (Fig. 6a), representing our membrane-bound structure. ATP-binding stabilizes the switch regions which in turn promotes the assembly[34]. The involvement of the G-interface also explains the strict ATP dependence of regular filament assembly on membranes (Supplementary Fig. 2)[23,25]. Accordingly, in the absence of nucleotide, EHDs can still interact with membranes (Supplementary Fig. 2)[23], but not assemble into a membrane-remodeling filament.

G-interface formation is associated with the displacement of the EH domain tail from the G-interface by the movement of the EH domains towards the periphery of the EHD filament (Supplementary

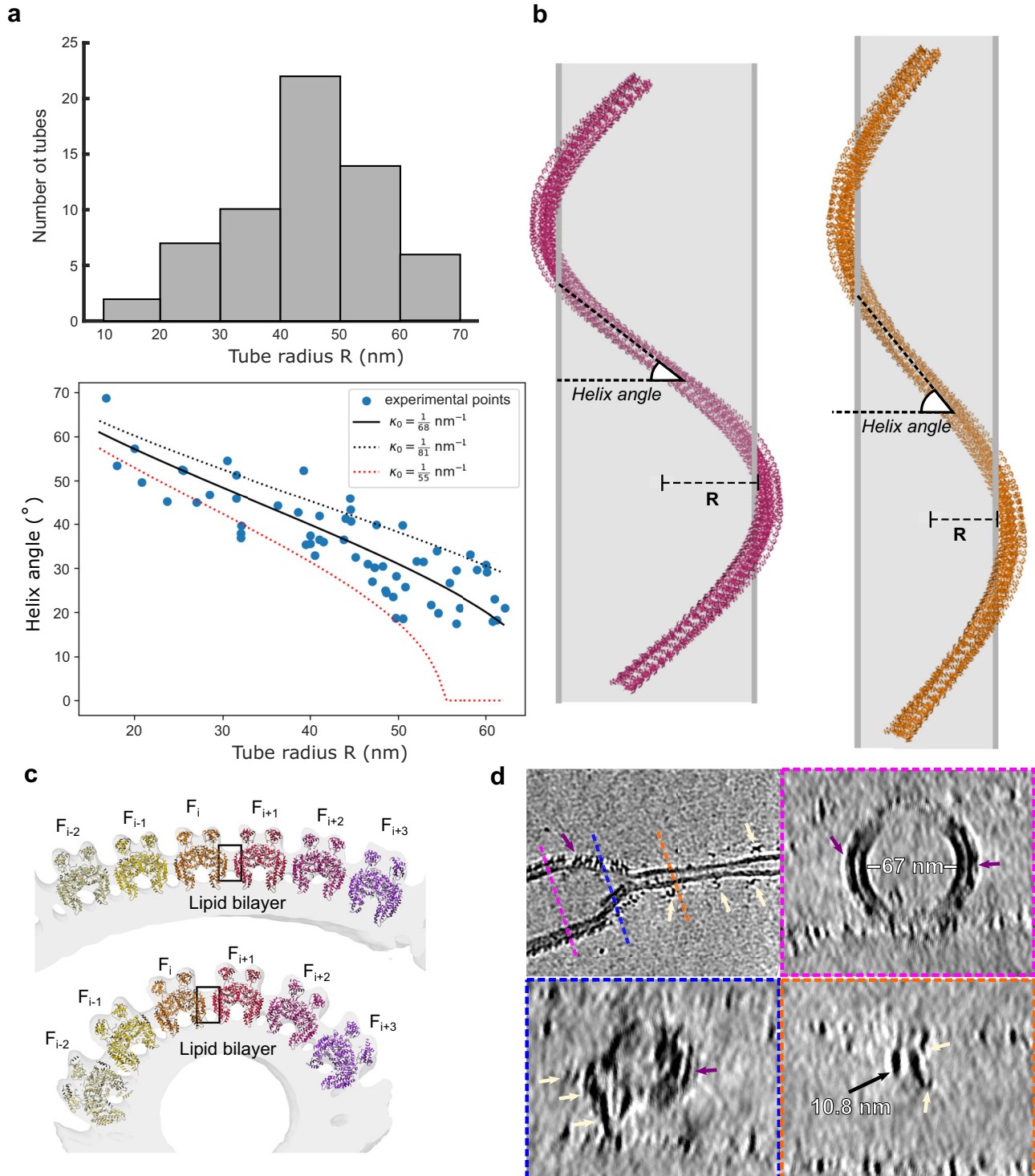

**Fig. 5 | EHD4 filaments adapt to the curvature of membrane tubes.**
**a** Distribution of tube radius (top) and the relation of tube radius and orientation of the filament along the tube axis (bottom). Solid line shows the best fitting spontaneous curvature (see Methods). Dotted lines show the limiting range to illustrate the confidence in predicting $\kappa_0$. **b** EHD4-covered membrane tubes show various assemblies. A single filament is depicted around the lipid tube (gray). The average helical assembly of the membrane-bound EHD4 structure is represented on the left and the tube with the smallest diameter is depicted on the right. Both filaments have 42 EHD4 dimers. **c** EHD4 filaments adopt different conformations in different tube diameters. The angle between filaments increases in tubes with higher curvature (bottom) along interface-4 (boxed) which acts as a hinge. **d** De-noised tomographic reconstruction of EHD4 oligomer on a thinning membrane. Cross sections along the tube are shown in magenta, blue and orange dashed boxes. Ordered EHD4 filaments form on a tube with a wide diameter (purple arrow), while membrane-bound EHD4 dimers (yellow arrows) were found on a narrow tube. Source data are provided as a Source Data file.

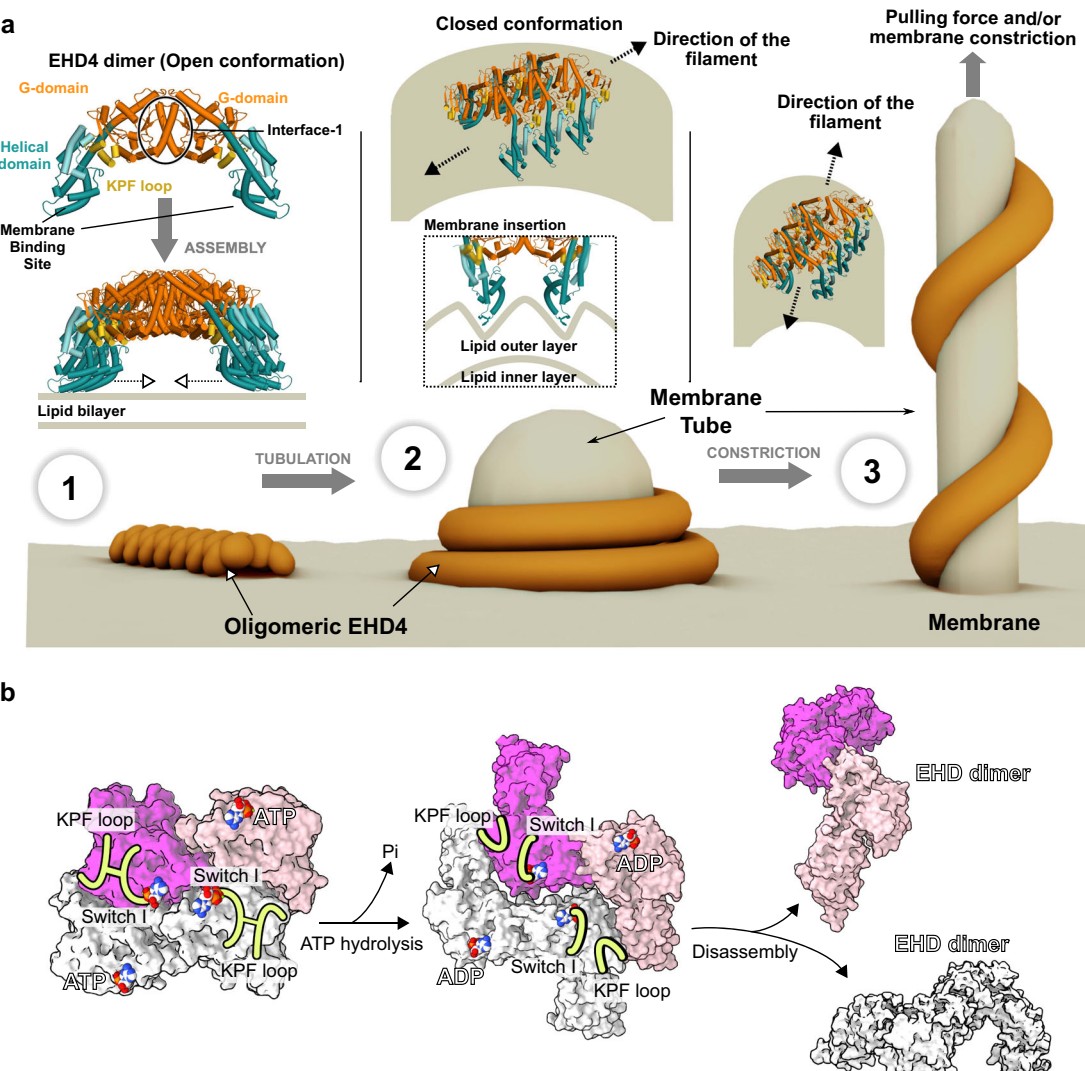

**Fig. 6 | Model for EHD-mediated membrane remodeling. a** 1) ATP-bound EHD dimers are recruited to flat membranes in the open conformation where they oligomerize via interface-2 into filaments of low curvature. 2) Membrane curvature induces the transition of the open to closed conformation. In turn, insertion of the membrane-binding site into the membrane promotes membrane curvature, which is associated with the formation of a stable helical filament via interface-3 (see also Supplementary Movies 3 and 4). Further constriction of the membrane tube will lead to an increase of the helical pitch. **b** ATP hydrolysis is expected to destabilize the G interface, leading to disassembly of the filament. The ADP-bound EHD dimer may convert back to the open conformation and dissociate from the membrane.

Fig. 6). In this orientation, the EH domains may bind to NPF-motif containing partner proteins (Supplementary Figs. 6e, f). Formation of the G-interface is accompanied by a stimulation of the slow ATP hydrolysis reaction in EHDs[21,22]. In this way, ATP hydrolysis may act as an intrinsic timer to disassemble the EHD scaffold: In the ADP-bound state, the switch regions are destabilized and the interaction of switch I with the KPF loop is reduced[32]. Accordingly, interfaces-2 and -3 are therefore weakened in the ADP-bound state, likely leading to dissociation of the oligomer (Fig. 6b). The ADP-bound EHD dimer may convert back to the open conformation and eventually dissociate from the membrane, therefore completing the ATPase cycle.

The EHD4 filament coat differs from the canonical oligomer architecture shared by several dynamin-related proteins, e.g. dynamin, DRP1, and Mgm1/OPA1. There, stalk interactions define the filament and nucleotide-dependent interactions between G-domains stabilize the inter-filament packing. EHDs on the other hand, orient the nucleotide-dependent interface along the filament direction and pack neighboring filaments with a stalk-stalk-like interaction. This change in architecture allows the intrinsic filament curvature to be modulated by nucleotide, or perhaps more interesting, the nucleotide state to be affected by the geometry of the filament. Additionally, whereas the pitch during active dynamin constriction is essentially fixed by the strong cross-filament interaction[53], EHD4 architecture may allow continuous reorientation of the filament as the underlying membrane curvature changes.

As the helix angle of the filament changes, the orientation of the individual EHD4 dimers adapts. This orientation has been previously suggested to be meaningful[22] as the membrane binding surface of EHD2 dimers in the closed conformation appears curved. Our structure shows that the dimer curvature is approximately 60° out of phase with the filament, meaning that when the helix angle is 60°, the dimer's curvature is aligned with the tube curvature. In this orientation, any curvature generation by the dimer should be maximized. Interestingly, the maximum helical angles observed are roughly 60°, where the tube radius is about 15 nm. This state, with a large pitch, may represent the maximum curvature that the EHD4 filament can stabilize. Further constriction of the tubes, for example by pulling forces and/or other membrane remodeling proteins[54], would be expected to destabilize the EHD filament on the membrane tube, leading to disassembly (Fig. 6b).

Previous mutagenesis data (Fig. 3)[22,32] are consistent with the formation of membrane-remodeling EHD filaments in the cell, although such scaffolds have not yet been unequivocally visualized in a cellular context. The low spontaneous membrane curvature of the EHD4 filament is in line with the reported cellular localization of EHD4 on macroendosomes[18] or early endosomes[55], which possess low membrane curvatures. In contrast to EHD4, ring-like assemblies around membrane tubes of higher curvatures were demonstrated for EHD1[25] and EHD2[22] (Supplementary Fig. 3), in agreement with their reported cellular localization on more highly curved cellular membranes, such as membrane tubes[5] and the neck of caveolae[14], respectively. Thus, the filaments' curvature preference of different EHD paralogues may be adapted to the architecture of the cellular membrane compartments they are acting on. Future super-resolution light microscopy and in situ electron microscopy studies are needed to characterize the detailed integration of the EHD filaments into their cellular membrane environment, their interaction with cellular partners and the cellular requirements for the formation and proposed ATPase-driven disassembly of the filaments.

Taken together, our structural analyses of the membrane-bound EHD4 scaffold elucidates novel insights into the coordination of the ATPase cycle with membrane recruitment, assembly and disassembly of the protein scaffold, and provides experimental insights into how membrane curvature is generated by EHD scaffolds.

## Methods

### Protein purification

Mouse EHD4 (residues 22-541, EHD4$^{\Delta N}$) and the indicated mutants, mouse EHD2 and EHD2$^{\Delta N}$ (residues 19-543)[22] were expressed from a modified pET28 vector as N-terminal His6-tag fusions followed by a PreScission protease cleavage site. Expression plasmids were transformed into E. coli host strain BL21(DE3)-Rosetta2 (Novagen). Cells were grown at 37 °C in TB medium, and protein expression was induced at an optical density of 0.5 by the addition of 40 µM isopropyl-β-D-thiogalactopyranoside (IPTG), followed by overnight incubation at 18 °C. Compared to EHD4$^{\Delta N}$, full-length EHD4 yields only insoluble protein with this expression approach. Upon centrifugation, cells were resuspended in resuspension buffer (50 mM Hepes/NaOH (pH 7.5), 500 mM NaCl, 25 mM imidazole, 2 mM MgCl$_2$, 2.5 mM β-mercaptoethanol (β-ME), 1 mM Pefabloc (Carl Roth), 1 µM DNase I (Roche)) and lysed in a microfluidizer. Following centrifugation (30,000 g, 1 h, 4 °C), cleared lysates were applied to a NiNTA column. The column was then extensively washed with washing buffer (50 mM Hepes/NaOH (pH 7.5), 700 mM NaCl, 10 mM CaCl$_2$, 1 mM ATP, 10 mM MgCl$_2$, 10 mM KCl) and afterwards with equilibration buffer (50 mM Hepes/NaOH (pH 7.5), 500 mM NaCl, 25 mM imidazole, 2 mM MgCl$_2$, 2.5 mM β-ME). The protein was eluted with elution buffer I (50 mM Hepes/NaOH (pH 7.5), 500 mM NaCl, 2 mM MgCl$_2$, 2.5 mM β-ME, and 300 mM imidazole). Following the addition of 150 µg PreScission protease per 5 mg of protein, the protein was dialyzed overnight against dialysis buffer (50 mM Hepes/NaOH (pH 7.5), 500 mM NaCl, 1 mM MgCl$_2$ and 2.5 mM β-ME). Following re-application of the protein to a NiNTA column to remove the His-tag, the protein was eluted with elution buffer II (50 mM Hepes/NaOH (pH 7.5), 500 mM NaCl, 2 mM MgCl$_2$, 2.5 mM β-ME, and 50 mM imidazole). The cleaved protein was concentrated using 30 kDa molecular weight cut-off concentrators (Amicon) and applied to a Superdex 200 gel filtration column equilibrated with SEC buffer (50 mM Hepes/NaOH (pH 7.5), 500 mM NaCl, 1 mM MgCl$_2$, and 2.5 mM β-ME). Fractions containing the EHD4 constructs were pooled, concentrated and flash-frozen in liquid nitrogen. The purified protein was nucleotide-free, as judged by High-Performance Liquid Chromatography (HPLC) analysis.

### Liposome preparation

Liposomes were prepared by adding 50 µL of 50% Folch extract from bovine brain fraction I, 40% phosphatidylethanolamine and 10% cholesterol to 200 µL of a chloroform/methanol (1:0.3 v/v) solution, which was then dried on a glass surface under an argon stream. Lipids were resuspended in liposome buffer (20 mM Hepes/NaOH (pH 7.5), 150 mM NaCl and 2.5 mM β-ME), sonicated in a water bath for 30 sec and extruded through a 1 µm filter.

### Tubulation assay

For membrane tubulation assays, 10 µM EHD4$^{\Delta N}$, EHD2 or EHD2$^{\Delta N}$ in tubulation buffer were incubated at room temperature for 20 min with 1 mg/ml liposomes in the absence or presence of 1 mM of the indicated nucleotide.

### ATPase assay

ATP hydrolysis of EHD4$^{\Delta N}$ was studied at 30 °C in 20 mM Hepes/NaOH (pH 7.5), 150 mM NaCl, 2.5 mM β-ME and 0.2 mM MgCl$_2$ using 10 µM protein and 100 µM ATP as a substrate, in the presence and absence of 1 mg/ml liposomes (see before). Reactions were started by adding the protein. At any given timepoint, aliquots from the reaction were 6 times diluted and flash-frozen in liquid nitrogen. Hydrolysis was measured using HPLC measurement. The nucleotides were separated with a reversed-phase column (C18 100 mm × 4.6 mm) and detected by absorption at 254 nm. Quantification was done by integrating the corresponding nucleotide peaks (see Source Data). GraphPad Prism 7.05 was used to plot the curve.

### Negative-stain EM

Liposomes at a concentration of 1 mg/ml were incubated for 20 min at room temperature with 10 µM protein in the presence of 1 mM ATP. Samples were applied for 30 sec to carbon-coated formvar films mounted on copper grids, excess liquid was removed, and the sample stained for 30 sec with 2% uranyl acetate. Four areas on three different grids were monitored on a Talos L120C (Thermo Fisher Scientific) at 120 kV equipped with a Ceta Detector to obtain representative images.

### Cryo-electron microscopy and image processing

Complexes formed of EHD4$^{\Delta N}$ and liposome were diluted with a buffer containing 10 nm colloidal gold. 4 µl of this mixture was applied on a glow-discharged Quantifoil R2/2 grid (Quantifoil Micro Tools GmbH) and flash-frozen in liquid ethane using a Vitrobot Mark II device (Thermo Fisher Scientific). The grids were stored under liquid nitrogen conditions until usage. Initial data was recorded on a Talos L120C microscope (Thermo Fisher Scientific) at 120 kV. The final set of 56 tilt series was collected on a Titan Krios G3 electron microscope (Thermo Fisher Scientific) operated at 300 kV equipped with a Gatan Quantum energy filter (slit with 20 eV) and a Gatan K2 detector using SerialEM 4.0 (ref. 56). The nominal magnification was 53,000× resulting in a pixel size of 2.628 Å. The data was acquired at a tilt range from −60 to 60 degrees using a dose-symmetric tilt scheme[57] at 3° increment. Tilt series were recorded as movies of 12 frames in counting mode and a dose rate of 2.3 e$^-$/Å$^2$ at defocus range of −3 to −6 um resulting in a total dose of 94 e$^-$/Å$^2$ per tilt series. Motion correction was carried out using MotionCor2 1.1.0 (ref. 58). The initial contrast transfer function defocus value for each image of the tilt series was estimated using CTFFind 4.1.10 (ref. 59). CTF correction was carried out by phase flipping using the program 'ctf phase flip' of the IMOD 4.11 software package[60]. Two copies of each tomogram were reconstructed using weighted back projection and seven rounds of the Simultaneous Iterative Reconstruction Technique (SIRT) in IMOD 4.11 (ref. 60).

For comparison, two grids of EHD4$^{\Delta N}$ in the apo form were prepared and data collected on a Titan Krios G3i transmission electron microscope (ThermoFisher Scientific, Server Version 2.15.3) operated at 300 kV equipped with an extra bright field-emission gun (XFEG), a BioQuantum post-column energy filter (Gatan) and a K3 direct electron detector (Gatan Digital Micrograph Version 3.32.2403.0). Images were recorded in low-dose mode as dose-fractionated movies using

SerialEM 4.0 in energy-filtered zeroloss (slit width 20 eV), nano-probe mode at a nominal magnification of 53,000x (resulting in a calibrated pixel size of 0.84 Å/px on the specimen level) in super-resolution mode with a 70 μM objective aperture.

## Subtomogram averaging

The workflow described uses a combination of Dynamo 1.1.157 and bespoke scripts from MATLAB version r2020b. Initial particle picking was done using a filament tracer by assigning the center of each tube. A first round of alignments was performed using SIRT-filtered reconstructed tomograms binned 3 times (7.884 Å/pixel). Particles for each tube were extracted using a box size of 128 pixels, randomized along its azimuth, averaged and low-pass filtered to 40 Å to generate the initial template for 3 rounds of coarse alignments. Oversampled particles converging onto the same coordinate were removed using Dynamo's separation in the tomogram parameter. Each tube was aligned individually and sub-boxed along the membrane to generate a section of the tube (Supplementary Fig. 4a). Averaged sections were merged, aligned to the template and low-pass filtered to 40 Å. Multi-reference analysis (MRA) was used to eliminate bad particles from the dataset. Next, CTF-corrected subtomograms were extracted and aligned to low-pass filtered references and the dataset was further divided into two half datasets, even and odd, for independent processing. Iterations were carried out starting from binned 3X data, using a low-pass filter of 20 Å, angular sampling of 12°, allowing shifts of 47 Å, and refinements were gradually improved by decreasing the binning factor, using less stringent low-pass filters and finer angular sampling. Final refinement steps were carried out on unbinned data extracted in 128 voxel boxes, using a low-pass filter set at 8 Å, angular sampling of 4°, and shift limits of 10 Å. A total of 23,813 subtomograms (11,906 and 11,907 for each half) contributed to the final average. Mask-corrected resolution assessment was carried out within the RELION 3.1 (ref. 61) postprocessing framework using a soft-edged mask around the central EHD4 tetramer, yielding a resolution of 7.6 Å at the 0.143 FSC cut-off (Supplementary Fig. 4). Local resolution estimation and local filtering were applied using Phenix Local anisotropic sharpening and Phenix Local resolution map in Phenix 1.20 (ref. 62). Figures were prepared using The PyMOL Molecular Graphics System, Version 2.5.0, Schrödinger, LLC., Chimera 1.14, Chimera X 1.1 (ref. 63) and Blender 3.1 (http://www.blender.org).

## Flexible fitting

The fitting procedure is summarized in Supplementary Movie 2. An atomic model consistent with the cryo-EM map was generated using MDfit with Gromacs 4.5.5[64]. MDfit uses the cryo-EM map as an umbrella potential to bias (i.e. deform) an underlying structure-based model (SBM)[65] in order to maximize the cross-correlation between the experimental density and the simulated electron density. An SBM is a molecular force field that is explicitly, albeit not rigidly, biased toward a certain native structure. The SBM for fitting was the EHD2 crystal structure (pdb code 4CID) with the sequence homology modeled by SWISS-MODELL[66] to that of EHD4 (residues 22-535). The portion of the SBM for the KPF loop (residue 114-137), which is missing from the EHD2 structure, is based on the EHD4 crystal structure (pdb code 5MVF). Building the SBM from the crystal structure ensured that the resulting model was maximally consistent with the crystal conformation. This entailed no significant changes in structure as the sequences are highly similar and included a missing loop in the crystal structure (residues 424–442). A preprocessing step was then necessary to move the EH domains within the dimer into a cis positioning because 4CID placed the EH domains in trans. This involved only the reorientation of the 424–442 loop, no other residue positions were changed. We refer to this dimeric structure as EHD4-init. Since the EH domain is missing, the SBM for the EH domain is generated from the EHD2 crystal structure (residues 443–538). An SBM using EHD4-init as the input structure was then generated using SMOGv2.3beta[65] with the template "SBM_AA" meaning all nonhydrogen atoms were explicitly represented.

The density corresponding to the central two dimers within the cryo-EM map was chosen as the constraint for MDfit, since this region had the best resolution. Relaxation of the SBM under the influence of the cryo-EM map is performed by molecular dynamics (MD), and, thus, requires an initial condition. Two EHD4-init were rigid body fit into the map using the "Fit in Map" tool of Chimera 1.14[63]. In order to compensate for the missing neighbors on either side of two dimers, the translational symmetry of the filament was exploited. Two additional copies of EHD4-init were added, positioned on either side, placed such that each dimer-dimer interface was identical. Technically, this was performed by 1) measuring the transformation X between the two central dimers in Visual Molecular Dynamics (VMD) 1.9.3, 2) duplicating the central dimers, and 3) applying X or -X to the duplicates. This four-dimer system served as the initial condition for MD. During MD, the duplicates were given strong position restraints, while the only constraint on the central dimers was the MDfit umbrella potential based on the cryo-EM map. Every $10^4$ MD steps, the duplicate dimers were repositioned. Through this iterative process, the structure converged within $3\times10^5$ steps. The middle two dimers were taken as the atomic model. Note that even though the filament's local C2 rotational symmetry was not explicitly enforced by us during MD, the fact that the SBM was based on a C2 symmetric structure ensured that this symmetry was included.

## Elastic model fitting

The fitted helical angle as a function of radius was defined by

$$\theta(r) \equiv \min\left[k_\kappa(\kappa-\kappa_0)^2 + k_\tau(\tau-\tau_0)^2 + k_\eta(\eta-90^\circ)^2\right] \quad (1)$$

where the bit in brackets is an elastic energy and the min returns $\theta$ such that the radius is $r$ and elastic energy is minimized. $\kappa$ is the curvature, $\tau$ is the twist, $\eta$ is the dimer orientation with respect to the tube axis, $\theta$ is the helix angle, and $r$ is the radius of the tube. $\kappa_0$ and $\tau_0$ are the spontaneous curvature and twist. Figure 5a (see also Source Data) shows the case for $k_\kappa=1, k_\tau=k_\eta=0$, where the twist elasticity and the dimer orientation are both negligible compared to filament curvature. Supplementary Fig. 7c shows two alternatives, the first with $k_\kappa=k_\tau=1, k_\eta=0$, which is typical for continuous filaments, and the second with $k_\kappa=1, k_\tau=0, k_\eta=0.2$, which accounts for a preference for the dimer curvature to align perpendicular to the tube axis. For a constant helix, $\kappa=\frac{r}{r^2+h^2}$, and $\tau=\frac{h}{r^2+h^2}$, and $\theta=\tan^{-1}\frac{h}{r}$, where $r$ is the radius and $2\pi h$ is the pitch. In our EHD4 filament, the dimer is oriented approximately 30° relative to the helix angle, defining $\eta=\theta+30^\circ$. Therefore, when $\eta=90^\circ\equiv\theta=60^\circ$, the dimer's footprint curvature is optimally oriented with respect to the membrane tube. The two best fits with $k_\eta=0$ are performed using the Python library scipy.optimize.least_squares 1.7.1 with $\kappa_0$ and $\tau_0$ as fitting parameters. The curve with $k_\eta=0.2$ uses $\kappa_0=\frac{1}{68}$ nm$^{-1}$ and has no free parameters to be fitted. Note that there is an analytic form for the minimum energy line with $k_\kappa=1, k_\tau=k_\eta=0$, which is given by

$$\theta(r) = \sqrt{(r\kappa_0)^{-1}-1} \quad (2)$$

## Reporting summary

Further information on research design is available in the Nature Portfolio Reporting Summary linked to this article.

## Data availability

The data that support this study are available from the corresponding authors upon reasonable request. The cryo-EM map has been deposited in the Electron Microscopy Data Bank (EMDB) under accession

code EMD-25362 (EHD4 filaments). Coordinates have been submitted to the Protein Data Bank (PDB) under accession code 7SOX (EHD4 asymmetric unit).

Coordinates of the previously published crystal structures used can be accessed via 4CID (EHD2) and 5MVF (EHD4$^{\Delta N}$). The source data underlying Fig. 5a and Supplementary Figs. 2b and 7c are provided as a Source Data file. Source data are provided with this paper.

## Code availability

Ad hoc dynamo scripts are available on Github: https://github.com/aamelo/dynamo_scripts [https://doi.org/10.5281/zenodo.7305315]. Angdist version 1.2 can be accessed via https://github.com/Guillawme/angdist/blob/main/README.md.

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

## Acknowledgements

This work was supported by grants from the Deutsche Forschungsgemeinschaft (SFB 958/A12 to O.D. and Z3 to C.S.), the ERC grant MitoShape (ERC-2013-CoG-616024 to O.D.), a Humboldt fellowship to J.K.N. and the iNEXT grant PID3536 VID5570. We would like to thank Wim Hagen for support during cryo-ET data collection at the EMBL cryo-EM facility and the Core Facility for cryo-Electron Microscopy (CFcryoEM) of the Charité - Universitätsmedizin Berlin for support in acquisition and analysis of data. The CFcryoEM was supported by the German Research Foundation (DFG) through grant No. INST 335/588-1 FUGG. We would like to thank Mikhail Kudryashev for advice and discussion.

## Author contributions

A.A.M. purified protein constructs, performed reconstitution of protein on membranes, optimized cryo-EM samples, processed data, determined the cryo-ET structure, analyzed the models and generated most of the figures and movies. T.S. screened cryo-EM samples and pre-processed data. J.K.N. performed model fittings and analyses and provided Supplementary Movie 2. E.V.-S. and C.H. cloned, purified protein constructs, and performed membrane-binding assays. E.V.-S. performed negative-stain experiments on EHD2. S.M. performed ATPase assays. J.L. assisted in data preprocessing. C.S. and O.D. supervised the project. A.M., J.K.N. and O.D. wrote the article, with input from all authors.

## Funding

## Competing interests

The authors declare no competing interests.
