## [Peer Review File · Nature Communications]

Cryo-electron tomography reveals structural insights into the membrane remodeling mode of dynamin-like EHD filamentsREVIEWER COMMENTS

Reviewer #1 (Remarks to the Author):

Melo et al present structural analysis of membrane bound EHD4. EHDs ATPases belong to the dynamin superfamily of proteins with potential functions in endosomal recycling, caveolar endocytosis and plasma membrane repair. In this context, structural insights into EHD self-assembly and disassembly on the membrane would provide tremendous insights into the functions of these proteins, which is exactly what this manuscript provides. This, combined with the fact that the structure explains previous results from mutagenesis, is a testament of its validity and importance.

I just have a few points for the authors to considered, which I hope would increase the appeal of this manuscript.

Why was the structure not solved for the full-length protein? An earlier paper from the same group had stated that purification of full-length EHD4 was not possible. But the authors could have tried the full-length EHD2 and thereby test the model that the N-terminus swings out of the G-domain upon membrane binding.

Can the authors clarify why is ATP binding required for membrane binding, when the helical domain is clearly exposed and accessible? Is this to do with avidity alone? Is the low affinity of the dimer per se incapable of recruiting EHDs to the membrane in the apo state? Seems like the model proposed in this manuscript favours this notion but is inconsistent with previous results from pelleting assays where binding was seen in the absence of nucleotides. Perhaps, inclusion of cryo-EM images of the protein with liposomes in presence of different nucleotides would be a nice inclusion to resolve this discrepancy.

Reviewer #2 (Remarks to the Author):

Dynamin-related proteins of the EHD family utilize their ATPase activity to remodel cellular membranes. Previous structural studies have revealed distinct conformational states of two isoforms, which - together with biochemical and cellular characterizations - provided the basis for a model for the proteins' nucleotide-dependent membrane remodeling activity. However, detailed insight into the membrane-bound EHD assemblies, depicting the proteins in a more native, functionally relevant environment, remained elusive.

In the present study, Melo and colleagues use cryo-electron tomography and sub-tomographic averaging to resolve EHD4 filaments bound to AMPPNP on lipid tubes. The resolution of the experimental data is sufficiently high to enable placement of high-resolution crystal structures, revealing novel interfaces contributing to the higher-order assembly and conformational changes within EHD4 monomers. In the filaments, EHD4 adopts the previously described closed state with intramolecular conformational changes arguing for a relief of autoinhibitory interactions involving the C-terminal protein-binding domain. Interestingly, there is some plasticity in the EHD4 filaments that allows the polymer to adopt to or stabilize a range of membrane tube diameters. Furthermore, the study confirms the previously proposed mechanism of membrane insertion of the protein's helical domain, which involves a set of hydrophobic residues that penetrate deeper into the membrane leaflet and positively charged residues that likely interact with the lipid headgroups. Observing this interaction experimentally is a rewarding feat. Together, this well-executed study describes a new structural snapshot of an important functional state of EHD4. The manuscript is complete, written and illustrated clearly in a way that should be accessible to a broader audience.

What is missing is an experimental correlation (or at least a discussion) of the structural assembly reconstituted in vitro with EHD4 oligomerization in cells. Although the functional effects of point mutations described in the literature can be correlated perfectly with the filament reconstruction

presented here, it is unclear whether such ordered arrays play a role in EHD4's cellular membrane remodeling activity. In addition, contrary to the last statement of the discussion, assembly and disassembly mechanisms were not addressed formerly in this study, since only stable filaments of AMPPNP-bound EHD4 were investigated. However, models for the processes leading up to and disassembling stable filaments were inferred from past analyses and the new insight presented here. But the models remained untested. Hence, a more nuanced summary of the key insights of this study would be warranted.

Minor points:

1. References to figure panels is not always chronological. For example, Supplemental Figure 2 is mentioned before Supplemental Figure 1C. Reorganizing the figure panels so that they can be introduced in order may improve readability.
2. Page 6, line 189: Since EHD4 has likely more than one conformational state on the membrane, it may be more appropriate to say "...good model for a membrane-bound EHD4 state." Alternatively, "...good model for the membrane-bound EHD4 determined in this study" could work as well.

Reviewer #3 (Remarks to the Author):

The manuscript by Melo et al, describe the structure of EHD4 bound to membrane utilizing cryo-electron tomography methods. The group achieved an impressive 7.6 Å resolution with this method and presents the first sub-nanometer structure of an EHD protein bound to membrane. When compared to previous crystal structures of EHD proteins, the membrane bound structure closely resembles the closed conformation. The reconstituted EHD4 decorated tubes used to determine the structure were highly variable in diameters and therefore not amenable to helical single particle methods. Instead subtomogram averaging was used to generate a structure in the range of 8 Å. The membrane-bound structure resembled the EHD2 crystal structure in the closed conformation. The initial model was built using rigid body fitting of each domain and then improved with the MDfit flexible fitting program to achieve the sub-nanometer resolution. From the structure, three main interfaces are defined that promote assemble of the oligomer. Compared to previous crystal structures the EH domains swing outward in the dimer and interact with neighboring dimers, which prevents the C-terminal tail from blocking the G domain interface. The tip of the helical domain inserts into the membrane as observed in the map and model in agreement with previous EPR measurements. Based on their membrane bound structure, the following model is proposed: The helical domain inserts into the membrane generating positive curvature, which may lead to the conversion from the open to the closed conformation and drive further oligomerization. Upon membrane binding the N-terminus reorients from the G domain pocket to the membrane allowing for the KPF loop to enter the pocket and create interface 2, stabilizing oligomerization. The EH domains also swing away from each other during oligomerization, allowing for neighboring G-domains to form interface 3 (stabilized by ATP-binding), a conserved interface conserved among dynamin-related proteins that induces increased hydrolysis. ATP hydrolysis would then lead to disassembly.

Comments:

Overall, this is a well-written manuscript describing the structure of EHD4 bound to membrane and proposes a model for coordinating membrane assembly with ATP hydrolysis.

Previous reports show formation of interface 3 leads to ATP hydrolysis stimulation. Include an ATPase assay with and without membrane showing the effects of lipid binding and potentially ATP hydrolysis stimulation.

Clarify the difference between twist stiffness, curvature stiffness and angle of dimer relative to the tube. An illustration of these terms would help.

In the discussion, add a section discussing how the model ties back to the biological function in caveolae and endosomal remodeling.

Minor comments:

Overall better labeling in figures to match the text would be helpful.

Label Interface-1 in supp fig 1B.

Label GPF motif in supp fig 1B

Line 99: list amino acids in "N-terminal sequence..."

Line 143: Hard to see the density associated with α -helices in supp Fig 4.

Line 156: "...monomers each (Fig. 1D,E and 2A)..."

Line 158: add aa "tryptophan (W238A)..."

Line 162: what are the aa's for the KPF loop?

Fig 2B: Label helix 8 and 12 in Fig 2b.

Supp Fig 4, interface 2, label KPF loop

We would like to thank the three reviewers for their overall positive and constructive assessment of our work. Based on your comments, we performed new experiments and modified the manuscript at several positions along your suggestions, as indicated in the detailed point-by-point response and marked in red in the manuscript. We also added the new Supplementary Movie 3 to more clearly represent the EHD4 assembly interfaces, and described the impact of the EHD4 filaments on bending the surface of the membrane tubes (p. 7). Finally, we shortened the title to comply with the format requirements of Nature Communications.

Reviewer #1 (Remarks to the Author):

Melo et al present structural analysis of membrane bound EHD4. EHDs ATPases belong to the dynamin superfamily of proteins with potential functions in endosomal recycling, caveolar endocytosis and plasma membrane repair. In this context, structural insights into EHD self-assembly and disassembly on the membrane would provide tremendous insights into the functions of these proteins, which is exactly what this manuscript provides. This, combined with the fact that the structure explains previous results from mutagenesis, is a testament of its validity and importance.

Thank you very much.

I just have a few points for the authors to considered, which I hope would increase the appeal of this manuscript.

Why was the structure not solved for the full-length protein? An earlier paper from the same group had stated that purification of full-length EHD4 was not possible. But the authors could have tried the full-length EHD2 and thereby test the model that the N-terminus swings out of the G-domain upon membrane binding.

Thank you for this suggestion. Indeed, full-length EHD4 cannot be purified in a soluble form, as previously stated in the Method section and now introduced in the result section. To still address this concern, we now provide new data in Supplementary Fig. 3, showing that EHD2 tubulates liposomes to much smaller diameters compared to EHD4, in the absence or presence of the N-terminus. Therefore, a comparison of EHD4^{ΔN} and EHD2 full length does not yield meaningful information on the role of the N-terminus in membrane remodeling. Along the referee's suggestion, we have spent already several years to obtain structures of membrane-bound EHD2 with and without the N-terminus. If structure solution is successful, these data will be better presented in a separate story, taking also into account the unique cellular location and function of EHD2 at the neck of caveolae.

Can the authors clarify why is ATP binding required for membrane binding, when the helical domain is clearly exposed and accessible? Is this to do with avidity alone? Is the low affinity of the dimer per se incapable of recruiting EHDs to the membrane in the apo state? Seems like the model proposed in this manuscript favours this notion but is inconsistent with previous results from pelleting assays where binding was seen in the absence of nucleotides. Perhaps, inclusion of cryo-EM images of the protein with liposomes in presence of different nucleotides would be a nice inclusion to resolve this discrepancy.

As suggested by the referee, we added new cryoEM data of EHD4^{ΔN} in the absence of nucleotide (new Supplementary Fig. 2A). These data confirm that EHD4^{ΔN} also binds to membranes in the nucleotide-free state, although apparently with reduced efficiency. Importantly and in agreement with previous data for EHD2, no regular protein coat is observed on the membranes in the absence of nucleotide, and membranes are not efficiently remodeled. Thus, in line with our model and the suggestion of the referee, ATP appears to promotes membrane binding by stimulating the formation of an oligomeric membrane-remodeling EHD scaffold. We added this idea to the discussion for clarification (p 10, top).

Reviewer #2 (Remarks to the Author):

Dynamamin-related proteins of the EHD family utilize their ATPase activity to remodel cellular membranes. Previous structural studies have revealed distinct conformational states of two isoforms, which - together with biochemical and cellular characterizations - provided the basis for a model for the proteins' nucleotide-dependent membrane remodeling activity. However, detailed insight into the membrane-bound EHD assemblies, depicting the proteins in a more native, functionally relevant environment, remained elusive.

In the present study, Melo and colleagues use cryo-electron tomography and sub-tomographic averaging to resolve EHD4 filaments bound to AMPPNP on lipid tubes. The resolution of the experimental data is sufficiently high to enable placement of high-resolution crystal structures, revealing novel interfaces contributing to the higher-order assembly and conformational changes within EHD4 monomers. In the filaments, EHD4 adopts the previously described closed state with intramolecular conformational changes arguing for a relief of autoinhibitory interactions involving the C-terminal protein-binding domain. Interestingly, there is some plasticity in the EHD4 filaments that allows the polymer to adopt to or stabilize a range of membrane tube diameters. Furthermore, the study confirms the previously proposed mechanism of membrane insertion of the protein's helical domain, which involves a set of hydrophobic residues that penetrate deeper into the membrane leaflet and positively charged residues that likely interact with the lipid headgroups. Observing this interaction experimentally is a rewarding feat. Together, this well-executed study describes a new structural snapshot of an important functional state of EHD4. The manuscript is complete, written and illustrated clearly in a way that should be accessible to a broader audience.

Thank you very much!

What is missing is an experimental correlation (or at least a discussion) of the structural assembly reconstituted in vitro with EHD4 oligomerization in cells. Although the functional effects of point mutations described in the literature can be correlated perfectly with the filament reconstruction presented here, it is unclear whether such ordered arrays play a role in EHD4's cellular membrane remodeling activity. In addition, contrary to the last statement of the discussion, assembly and disassembly mechanisms were not addressed formerly in this study, since only stable filaments of AMPPNP-bound EHD4 were investigated. However, models for the processes leading up to and disassembling stable filaments were inferred from past analyses and the new insight presented here. But the models remained untested. Hence, a more nuanced summary of the key insights of this study would be warranted.

The relation of our data to the cellular function and cellular architecture of EHD filaments is indeed an important issue. As mentioned by the referee, previous mutagenesis data are consistent with the formation of cellular EHD scaffolds, but these scaffolds have so far not been experimentally visualized in their cellular context. We also agree that filament disassembly has not been experimentally addressed in our study and toned down the corresponding conclusions. Furthermore, we rewrote the final paragraph of the discussion, focusing specifically on the new cellular insights on EHD scaffolds our study conveys, including the determined curvature preference, the open questions in the field and how they could be addressed in future studies (p 11).

Minor points:

1. References to figure panels is not always chronological. For example, Supplemental Figure 2 is mentioned before Supplemental Figure 1C. Reorganizing the figure panels so that they can be introduced in order may improve readability.

Thanks, this was corrected as suggested. Old Supplementary Fig. 1C is now new Supplementary Fig. 4A so that the order of the figures is streamlined across the manuscript.

2. Page 6, line 189: Since EHD4 has likely more than one conformational state on the membrane, it may be more appropriate to say "...good model for a membrane-bound EHD4 state." Alternatively, "...good model for the membrane-bound EHD4 determined in this study" could work as well.

Thanks, changed as suggested.

Reviewer #3 (Remarks to the Author):

The manuscript by Melo et al, describe the structure of EHD4 bound to membrane utilizing cryo-electron tomography methods. The group achieved an impressive 7.6 Å resolution with this method and presents the first sub-nanometer structure of an EHD protein bound to membrane. When compared to previous crystal structures of EHD proteins, the membrane bound structure closely resembles the closed conformation. The reconstituted EHD4 decorated tubes used to determine the structure were highly variable in diameters and therefore not amenable to helical single particle methods. Instead subtomogram averaging was used to generate a structure in the range of 8 Å. The membrane-bound structure resembled the EHD2 crystal structure in the closed conformation. The initial model was built using rigid body fitting of each domain and then improved with the MDfit flexible fitting program to achieve the sub-nanometer resolution. From the structure, three main interfaces are defined that promote assemble of the oligomer. Compared to previous crystal structures the EH domains swing outward in the dimer and interact with neighboring dimers, which prevents the C-terminal tail from blocking the G domain interface. The tip of the helical domain inserts into the membrane as observed in the map and model in agreement with previous EPR measurements. Based on their membrane bound structure, the following model is proposed: The helical domain inserts into the membrane generating positive curvature, which may lead to the conversion from the open to the closed conformation and drive further oligomerization. Upon membrane binding the N-terminus reorients from the G domain pocket to the membrane allowing for the KPF loop to enter the pocket and create interface 2, stabilizing oligomerization. The EH domains also swing away from each other during oligomerization, allowing for neighboring G-domains to form interface 3 (stabilized by ATP-binding), a conserved interface conserved among dynamin-related proteins that induces increased hydrolysis. ATP hydrolysis would then lead to disassembly.

Comments:

Overall, this is a well-written manuscript describing the structure of EHD4 bound to membrane and proposes a model for coordinating membrane assembly with ATP hydrolysis.

Thank you!

Previous reports show formation of interface 3 leads to ATP hydrolysis stimulation. Include an ATPase assay with and without membrane showing the effects of lipid binding and potentially ATP hydrolysis stimulation.

Thanks for this suggestion. We now added new experiments showing that the ATPase activity of EHD4^{ΔN} is also stimulated by liposomes composed of the membrane mixture used for the reconstruction (Supplementary Fig. 2B). These experiments were carried out by Saif Mohd who is now a new co-author.

Clarify the difference between twist stiffness, curvature stiffness and angle of dimer relative to the tube. An illustration of these terms would help.

This information is now provided in the new Supplementary Fig. 7A and 7B.

In the discussion, add a section discussing how the model ties back to the biological function in caveolae and endosomal remodeling.

As also requested by referee 2, we added a final paragraph to the discussion describing the impact of our structural work to understand the cellular function of different EHD scaffolds (p 11).

Minor comments:

Overall better labeling in figures to match the text would be helpful.

Label Interface-1 in supp fig 1B.

Label GPF motif in supp fig 1B

Thanks for the hints, the figure has been modified accordingly.

Line 99: list amino acids in "N-terminal sequence..."

We added the sequence of the N-terminal residues to the new Supplementary Fig. 1C.

Line 143: Hard to see the density associated with α -helices in supp Fig 4.

This figure has been revised accordingly (see Supplementary Fig. 5).

Line 156: "...monomers each (Fig. 1D,E and 2A)..."

The additional figure link was added.

Line 158: add aa "tryptophan (W238A)..."

Added as suggested.

Line 162: what are the aa's for the KPF loop?

We added this information as a sequence alignment to Supp. Fig. 1C.

Fig 2B: Label helix 8 and 12 in Fig 2b.

Done as suggested.

Supp Fig 4, interface 2, label KPF loop

Done as suggested (see new Supplementary Fig. 5).

REVIEWERS' COMMENTS

Reviewer #1 (Remarks to the Author):

The authors have adequately addressed my queries. I have no further comments and congratulate the authors on this wonderful work.

Reviewer #2 (Remarks to the Author):

The authors have addressed fully my comments and questions. Congratulations to this insightful study!

Reviewer #3 (Remarks to the Author):

I am satisfied with the authors revised manuscript and my concerns have all been addressed.